# The Influence of the Exclusion of Central Necrosis on [^18^F]FDG PET Radiomic Analysis

**DOI:** 10.3390/diagnostics11071296

**Published:** 2021-07-19

**Authors:** Wyanne A. Noortman, Dennis Vriens, Charlotte D. Y. Mooij, Cornelis H. Slump, Erik H. Aarntzen, Anouk van Berkel, Henri J. L. M. Timmers, Johan Bussink, Tineke W. H. Meijer, Lioe-Fee de Geus-Oei, Floris H. P. van Velden

**Affiliations:** 1Section of Nuclear Medicine, Department of Radiology, Leiden University Medical Center, 2333 ZA Leiden, The Netherlands; d.vriens@lumc.nl (D.V.); c.d.y.mooij@lumc.nl (C.D.Y.M.); l.f.de_geus-oei@lumc.nl (L.-F.d.G.-O.); f.h.p.van_velden@lumc.nl (F.H.P.v.V.); 2TechMed Centre, University of Twente, 7522 NB Enschede, The Netherlands; c.h.slump@utwente.nl; 3Technical Medicine, Delft University of Technology, 2628 CD Delft, The Netherlands; 4Department of Radiology and Nuclear Medicine, Radboud University Medical Center, 6525 GA Nijmegen, The Netherlands; erik.aarntzen@radboudumc.nl; 5Division of Endocrinology, Department of Internal Medicine, Radboud University Medical Center, 6525 GA Nijmegen, The Netherlands; anouk.vanberkel@radboudumc.nl (A.v.B.); henri.timmers@radboudumc.nl (H.J.L.M.T.); 6Radiotherapy and OncoImmunology Laboratory, Department of Radiation Oncology, Radboud University Medical Center, 6525 GA Nijmegen, The Netherlands; jan.bussink@radboudumc.nl; 7Department of Radiation Oncology, University Medical Center Groningen, 9713 GZ Groningen, The Netherlands; t.van.zon@umcg.nl

**Keywords:** radiomics, [^18^F]FDG PET/CT, tumour delineation, central necrosis

## Abstract

Background: Central necrosis can be detected on [^18^F]FDG PET/CT as a region with little to no tracer uptake. Currently, there is no consensus regarding the inclusion of regions of central necrosis during volume of interest (VOI) delineation for radiomic analysis. The aim of this study was to assess how central necrosis affects radiomic analysis in PET. Methods: Forty-three patients, either with non-small cell lung carcinomas (NSCLC, *n* = 12) or with pheochromocytomas or paragangliomas (PPGL, *n* = 31), were included retrospectively. VOIs were delineated with and without central necrosis. From all VOIs, 105 radiomic features were extracted. Differences in radiomic features between delineation methods were assessed using a paired *t*-test with Benjamini–Hochberg multiple testing correction. In the PPGL cohort, performances of the radiomic models to predict the noradrenergic biochemical profile were assessed by comparing the areas under the receiver operating characteristic curve (AUC) for both delineation methods. Results: At least 65% of the features showed significant differences between VOI_vital-tumour_ and VOI_gross-tumour_ (65%, 79% and 82% for the NSCLC, PPGL and combined cohort, respectively). The AUCs of the radiomic models were not significantly different between delineation methods. Conclusion: In both tumour types, almost two-third of the features were affected, demonstrating that the impact of whether or not to include central necrosis in the VOI on the radiomic feature values is significant. Nevertheless, predictive performances of both delineation methods were comparable. We recommend that radiomic studies should report whether or not central necrosis was included during delineation.

## 1. Introduction

Tumour morphology might be heterogeneous with alternating regions of relatively vital tumour tissue, mild hypoxia, severe hypoxia and necrosis [1]. Tumour hypoxia manifests itself predominantly in solid tumours [2]. Central necrosis of tumours occurs as a result of hypoxia and is caused by uncontrolled oncogene-driven proliferation without efficient vasculature, inducing a nutrient and oxygen shortage [1]. As a morphological marker, central necrosis is associated with poor prognosis in a variety of cancers [3,4,5], including non-small cell lung carcinomas (NSCLC) [6,7]. Larger regions of necrosis can be detected on 2-[^18^F]fluoro-2-deoxy-D-glucose positron emission tomography ([^18^F]FDG PET/CT) as an often centrally located region with little to no tracer uptake.

Radiomics aims to quantify the geometry and tracer uptake, including uptake heterogeneity, of tumours by using first order, shape and texture features and hypothesizing that these features can be used for tumour characterisation, prognostic stratification and response prediction in precision medicine [8]. It is uncertain whether regions of central necrosis should be added to the delineation of the tumour, since the effect of delineation methods on the predictive value of the radiomic signature remains unknown [9]. Semi-automatic tumour delineation methods used in radiomic analysis apply isocontours by using fixed or adaptive thresholds [10,11] or more advanced algorithms, such as the fuzzy locally adaptive Bayesian (FLAB) algorithm [12]. Although these methods have shown to be highly reproducible [10,13], they often underestimate the true (anatomical) tumour volume by excluding (up to a certain degree) regions of low tracer uptake. Some studies manually add the excluded regions of low tracer uptake to the volume of interest (VOI), but this is not always clearly reported. It is hypothesised that the addition of a region of central necrosis to the VOI may influence all three radiomic feature classes numerically. First order features might be affected by the addition of voxels with low grey levels and this skews the intensity histogram. Shape features might be influenced by the different 3D morphology of the VOI when central necrosis is included. Texture features, representing spatial relationships between voxels in terms of run lengths or size zones of the same voxel values or combinations of neighbouring voxel values, might change as well. The introduction of an area of central necrosis might, for instance, result in long runs with low values that will change the run length matrix and, as a result, the feature values. The Image Biomarker Standardisation Initiative (IBSI), which is an independent international collaboration working towards standardising the extraction of image biomarkers, provides reporting guidelines for radiomic studies but, up to this point, does not specify the need to report on the inclusion/exclusion of necrosis while describing the used segmentation method [14]. Moreover, the effect of the delineation method, including whether or not to include central necrosis, on the performance of the radiomic signature for predicting underlying tumour biology remains unknown [9].

This study explores how central necrosis influences PET radiomic analysis by assessing the differences in radiomic features and the predictive performance of features extracted from VOIs delineated using an isocontour method with and without the manual addition of the region of central necrosis for two datasets of NSCLC and pheochromocytomas or paragangliomas (PPGL), catecholamine-producing neuroendocrine tumours that arise from the chromaffin cells of the adrenal medulla and extra-adrenal sympathetic paraganglia [15].

## 2. Materials and Methods

### 2.1. Patient Population, Data Acquisition and Image Reconstruction

Subjects from two cohorts of patients who underwent an [^18^F]FDG PET/CT in a single academic centre were retrospectively included to study the effect of different aspects of central necrosis on the radiomic analysis. A cohort of patients with non-small cell lung-carcinomas (NSCLC, *n* = 35), generally presenting a high tumour-to-background ratio, and a cohort of patients with pheochromocytomas or paragangliomas (PPGL, *n* = 77), generally presenting a low tumour-to-background ratio, were included.

The NSCLC cohort is a previously published prospective cohort [16]. Patients underwent a dynamic [^18^F]FDG PET/CT scan with the primary tumour located centrally in the field of view using the Biograph Duo or Biograph 40 mCT (Siemens Healthineers, Erlangen, Germany) at the Radboud University Medical Center between 2009 and 2014. Only tumours with a diameter larger than 30 mm were included to minimise the influence of partial volume effects and to be able to reliably quantify uptake heterogeneity [17]. Imaging was in accordance with European Association of Nuclear Medicine (EANM) guidelines for tumour PET imaging [18]. Patients fasted for at least 6 h before imaging and serum glucose levels were below 8 mmol/L. Directly after the start of the acquisition, a standardised infusion of 3.45 MBq of [^18^F]FDG per kilogram of body weight started. The final time frame (50–60 min p.i.) of the dynamic series was used in the current study. Voxel sizes were 2.56 × 2.56 × 3.38 and 1.59 × 1.59 × 2.03 mm^3^ for the Biograph Duo PET/CT and Biograph 40 mCT PET/CT, respectively. This study has been reviewed and approved by the Commission on Medical Research Involving Human Subjects Region Arnhem-Nijmegen, the Netherlands. All patients signed an informed consent form.

The PPGL patients who underwent a [^18^F]FDG PET/CT scan in the Radboud University Medical Center between 2011 and 2018 were retrospectively included. A selection of these patients has previously been described [15,19]. Static PET/CT images were acquired using the Biograph 40 mCT (Siemens Healthineers, Erlangen, Germany), in accordance with aforementioned EANM guidelines [18]. Patients fasted for at least 6 h and serum glucose levels were below 8 mmol/L. Image acquisition (3 or 4 min per bed position) started 60 (55–75) minutes after intravenous administration of [^18^F]FDG (dosage according to a non-linear dosage regimen based on body weight; details can be found in Appendix A). The reconstructed voxel size was 3.18 × 3.18 × 3.00 mm^3^. This retrospective database study has been reviewed and approved by the Commission on Medical Research Involving Human Subjects Region Arnhem-Nijmegen, the Netherlands. Informed consent was waived due to the retrospective nature of the study. Patients that objected to the use of their anonymised data were excluded.

Additional details on patient preparation, data acquisition, image reconstruction, image processing and radiomic analysis can be found in Appendix A: the IBSI reporting guidelines [14].

### 2.2. Image Analysis

#### 2.2.1. Image Processing

For NSCLC, images were interpolated to isotropic voxels of 3.38 × 3.38 × 3.38 mm^3^ using trilinear interpolation with the grids aligned by the centre using MATLAB version 2017b (Mathworks, Natick, MA, USA) [14]. PPGL images were not interpolated since the voxels were almost isotropic (3.18 × 3.18 × 3.00 mm^3^).

#### 2.2.2. Volumes of Interest Delineation

VOIs were delineated semi-automatically using 3DSlicer version 4.11 (www.slicer.org, accessed on 1 March 2021) [20] and in-house built software implemented in Python version 3.7 (Python Software Foundation, Wilmington, Delaware). The often peripheral region of the tumour showing increased [^18^F]FDG uptake (VOI_vital-tumour_) was delineated using a semi-automatic threshold-based method and corrected for local background [10]. A threshold of 41% of the peak standardised uptake value (SUV_peak_) that was obtained using a sphere of 12 mm diameter [21] was selected, since the delineated tumour sizes of this method agreed best with pathological tumour sizes [22]. As tumours in the PPGL cohort showed low contrast between the tumour and surrounding tissue, boxing was applied to exclude the surrounding [^18^F]FDG-avid tissues. VOI_gross-tumour_ was generated by manually adding the volumes of central necrosis to VOI_vital-tumour_, using the low-dose CT as a visual reference. The necrotic tumour fraction (NTF) was determined using Equation (1):NTF = 1 − VOI_vital-tumour_/VOI_gross-tumour_(1)
where a NTF of 0 indicates no central necrosis and a higher NTF indicates larger volumes of necrosis. Patients were selected when NTF > 0.

#### 2.2.3. Radiomic Feature Extraction

For each selected patient, 105 radiomic features were extracted from VOI_vital-tumour_ and VOI_gross-tumour_ using PyRadiomics version 3.0 in Python version 3.7 (Python Software Foundation, Wilmington, DE, USA) [23]: 18 first order features, 14 shape features, 22 grey level cooccurrence matrix (GLCM) features, 16 grey level run length matrix (GLRLM) features, 16 grey level size zone matrix (GLSZM) features, 14 grey level dependence matrix (GLDM) features and 5 neighbouring grey tone difference matrix (NGTDM) features. A fixed bin size of 0.5 g/mL was applied.

### 2.3. Statistical Analysis

Statistical analyses were performed in SPSS version 25 (IBM Statistics, Chicago, IL, USA). Per cohort and for both cohorts together, differences in radiomic features extracted from VOI_vital-tumour_ and VOI_gross-tumour_ were assessed using a Wilcoxon signed-rank test or a paired t-test after testing for (log-)normality. Since over one hundred features are tested simultaneously, some features may show a significant difference between both delineation methods by chance, which increases the false discovery rate [24]. Therefore, the Benjamini–Hochberg multiple testing correction was performed [25]. The Benjamini–Hochberg correction determines the significance level for specific feature (*p_i_*) using Equation (2):(2)$$p_{i}<\left(\frac{i}{n}\right)a$$
where *i* is the ranking of a feature when ranking all features based on the significance level of the paired *t*-test from smallest to largest, *n* is the total number of features and *a* is the original significance level (*a* = 0.05). Additional subset analyses of all patients based on the NTF and SUV_max_ were performed by creating three equally-sized groups for low, medium and high values: NTF: ≤0.12, 0.12 < NTF ≤ 0.36, >0.36; SUV_max_: ≤4.61, 4.61 < SUV_max_ ≤ 12.09, >12.09 g/mL. Differences in numbers of affected features per cohort and subgroup were assessed using the Fisher’s exact test. Overlaps in the affected features per cohort and subgroup were visualised using Venn diagrams.

For the PPGL cohort, the predictive performance for the underlying tumour biology of the radiomic models based on features derived from the different delineation methods was assessed by binary logistic regression in R version 3.6.0 (R Foundation for Statistical Computing, Vienna, Austria). Moreover, a radiomic model was created out of features from both delineation methods, assuming that both features contain different information. The response variable in regression was the noradrenergic biochemical profile of the PPGLs. Unsupervised feature selection or dimension reduction was performed to deal with multicollinearity and high dimensionality, which occurs when the number of features largely exceeds the number of patients. As a rule of thumb, 1 feature was selected for every 10 subjects [26] and 3 features were selected to be tested (PPGL dataset: *n* = 31 patients). The predictive performance of the radiomic models was not assessed for the NSCLC cohort since this cohort consisted of only 12 patients, which corresponds to only 1 feature to be tested and is inadequate to explain sufficient variance of the dataset. Dimension reduction in the PPGL dataset using redundancy filtering and factor analysis was performed using the FMradio (Factor Modeling for Radiomics Data) R-package version 1.1.1 [27]. Features were scaled (centred around 0, variance of 1), avoiding that features with the largest scale dominated the analysis. Redundancy filtering of the Pearson correlation matrix of features is performed with a threshold of τ = 0.95 and, from each group, one feature is retained. Factor analysis of the redundancy filtered correlation matrix with an orthogonal rotation was executed so that the first factor explained the largest possible variance in the dataset; the succeeding factors explained the largest variance in orthogonal directions. The sampling adequacy of the model, which is quantified by the Kaiser-Meier-Olkin (KMO) statistic, was predefined to be ≥0.9. The feature with the highest loading on a single factor was selected for regression analysis. The three selected features are associated with the noradrenergic biochemical profile using multiple binary logistic regression. Areas under the curve (AUC) of the receiver operating characteristic (ROC) of the radiomic models based on the three selected features for VOI_vital-tumour_, VOI_gross-tumour_ and combined were computed and compared using DeLong’s test for paired ROC curves. A sham experiment was conducted to validate the findings by randomisation of the outcome labels (noradrenergic biochemical profile) [28]. This takes into account the prevalence of the outcome and the distributions and multicollinearity of the radiomic features but uncouples their hypothesised relation. Binary logistic regression was performed and the sham experiment was repeated 100 times to calculate the mean AUCs.

## 3. Results

Patient characteristics of included patients with central necrosis (*n* = 43) are presented in Table 1. In the NSCLC cohort, central necrosis was observed in 12 out of 35 patients (34%). In the PPGL cohort, central necrosis was observed in 31 of 77 patients (40%; Figure 1).

At least 65% of the features were affected by the choice of delineation (Table 2). The PPGL population was influenced the most with 79% of affected features, compared to 65% in the NSCLC population, which is a significant difference between the populations (*p* = 0.031). Out of the 105 features, 61% were affected in the NSCLC cohort as well as the PPGL cohort. For all patients taken together, even 82% of the features were affected.

First order features were affected substantially, with at least 72% of features, followed by texture features with at least 66% of features. Shape features were affected the least, with 50% of affected features in both datasets. Of all texture feature classes, GLCM features were affected the least, with a maximum of 68% of features. For all other classes, the maximum number of affected features was at least 80%.

The size of the NTF appeared to influence the number of affected shape features (nonsignificant; Table 3). For a small NTF, 36% of the features were affected, increasing to 57% and 71% for medium and large NTFs, respectively. Moreover, for small and medium NTFs 100% of the NGTDM features were affected compared to only 40% for a large NTF.

Although nonsignificant, the value of the SUV_max_ appeared to influence the number of affected features (Table 4) and a higher value resulted in more affected features (56%, 66% and 70% for low, medium and high values, respectively). This increasing trend could also be observed for all texture feature classes except for GLCM features, where the number of affected features decreased with an increasing SUV_max_.

Overlap in affected features between cohorts and subgroups is generally high (Figure 2), yet the affected shape features varied largely between cohorts and for different-sized NTFs. It can be observed that many features that were affected in one subset or cohort, were also affected in most of the other subsets or cohorts (Appendix A).

For each of the three radiomic models evaluating predictive performance (VOI_vital-tumour_, VOI_gross-tumour_ and combined) for the PPGL dataset, three factors were retained and the three best corresponding features were selected (Table 5). The KMOs of the models were excellent (>0.96). AUCs varied 0.791–0.829, but were not significantly different between the radiomic models (VOI_vital-tumour_ vs. VOI_gross-tumour_: *p* = 0.775; VOI_vital-tumour_ vs. combined: *p* = 0.625; VOI_gross-tumour_ vs. combined: *p* = 0.874; Figure 3). The mean AUCs of the sham experiments were lower (0.645–0.655) than the AUCs of the radiomic models, indicating the validity of the findings.

## 4. Discussion

In this study, we assessed the effect of the inclusion of central necrosis during tumour delineation on radiomic analysis in two cohorts of patients with NSCLC and PPGL. Around two-third of radiomic features showed significant differences between adaptive threshold delineation with and without manual addition of the region of central necrosis. Nevertheless, the predictive performance of radiomic models with and without central necrosis for the noradrenergic biochemical profile of PPGLs was not significantly different. Due to the low number of subjects, the predictive performance was not assessed for the NSCLC cohort.

At least 65% of all features were significantly affected after adjustment for multiple-testing by the difference in delineation method. Less features were affected in the NSCLC cohort compared to the PPGL cohort (65% versus 82%, respectively), which is likely a result of lower power of the test due to a smaller cohort (12 versus 31, respectively).

More than 72% of the first order features, describing the distribution of voxel intensities in a histogram, significantly changed when central necrosis, i.e., lower intensity values, was added to the VOI. The number of affected first order features increases with a higher SUV_max_, which can be explained by the larger range of voxel values in the intensity histogram in the case of a high SUV_max_ after the addition of a region with central necrosis.

At least 66% of the texture features, describing the spatial relationships between individual voxels in terms of run lengths, size zones of the same voxel values or combinations of neighbouring voxel values, were affected by central necrosis, which resulted in a change of the spatial relationships between the voxels. It is beyond the scope of this study to dive into the mathematical definition of all texture features, but we will highlight some of our findings and possible explanations.

Similarly to first order features, the number of affected texture features also increased with increasing SUV_max_. The introduction of a region with low grey levels might result in longer run lengths and size zones with low values. In a tumour with relatively high grey levels, this might result in a larger run length or size zone matrix, with high incidences for the low grey values (central necrosis) and for the high grey levels (edge of the tumour) and low incidences in the middle range. For a tumour with a lower SUV_max_, the matrices remain smaller, with incidences in the low and middle ranges and this results in different feature values.

Furthermore, it is remarkable that almost all normalised texture features were significantly different for both delineation methods for the NSCLC cohort, the PPGL cohort and all patients combined: The *normalised* GLCM inverse difference and inverse difference moment, the GLRLM grey level non-uniformity and run length non-uniformity and the GLSZM grey level non-uniformity and size zone non-uniformity were significantly different between both delineation methods. The normalised GLDM dependence non-uniformity was only different in the PPGL cohort and the combined cohort. Normalisation of GLCM features is performed to improve classification accuracy [29]. Normalised features are standardised for the number of elements in their respective matrix, i.e., the GLCM consists of the square of the number of discretised grey levels and the GLRLM consist of the product of the number of discretised grey levels and the maximal run length [14]. Since the number of discretised grey levels and the maximal run length increase by the addition of the region of central necrosis, it could result in a decrease in feature values, resulting in differences in feature values between delineation methods.

Compared to other features classes, shape features were affected least frequently by the choice of the delineation method, but 50% of shape features were still affected. Some shape features consider the outer diameter or morphology of the VOI, which, in most cases, did not expressively change when adding the region of central necrosis. Nevertheless, in some cases the region of central necrosis touched the outer surface of the volume of interest (3D U-shape) and caused some features to change. The number of affected shape features increased with the NTF as a result of a larger additional region of necrosis.

Several studies on repeatability and reproducibility showed that radiomic feature values are affected by delineation methods [30]. Unfortunately, overviews of repeatability and reproducibility on a feature level are scarce and are often limited to feature classes. Traverso et al. wrote a systematic review on repeatability and reproducibility of radiomic features and assessed to what extent (highly likely/probable/less likely) the different feature classes were affected by different processing steps of the radiomic pipeline [31]. They found that it is probable that semi-automatic VOI delineation exerts an adverse effect on repeatability and reproducibility of texture features. Moreover, in the case of shape features, this adverse effect is probable but when compared to shape features derived from CT, PET shape features are more reproducible. According to Traverso et al., first order features are less likely to be affected by the delineation method, which is in sharp contrast with our study that shows that first order features are affected in particular by the delineation method. They also present that entropy was consistent among the most repeatable and reproducible first order features [31]. However, in our study it can be observed that entropy is one of the features that is significantly affected by the delineation method in all cohorts and subgroups. Coarseness and contrast (GLCM as well as NGTDM), on the other hand, are considered among the least reproducible features [31,32], whereas our results show that coarseness and GLCM contrast are affected in only two and zero out of nine subgroups (Appendix A), respectively. This shows that non-repeatable or non-reproducible features are not the only features affected by central necrosis and therefore the choice of the delineation method concerning central necrosis should be considered in the design of radiomic studies.

Our study analysed the differences in radiomic features in two cohorts with different tumour types, showing a different tumour-to-background ratio. While the same image analysis (VOI delineation and radiomic feature extraction) was performed, acquisition and reconstruction settings were different between cohorts even though both protocols were in accordance with the EANM guidelines. The features affected by the delineation method, however, were highly similar between both cohorts, indicating that the effect of the delineation method concerning central necrosis is independent of the tumour type. Therefore, whether or not to include central necrosis in the tumour delineation is an important factor to consider when performing clinical radiomic studies. We hypothesise that this might also apply to other tumour types, but the number of affected features might vary as a result of tumour characteristics such as the tracer uptake and distribution, tumour geometry and NTF.

While almost two-third of radiomic feature values were significantly affected by the choice of delineation method, the predictive performances of the radiomic models, as assessed in the PPGL cohort, were not affected accordingly. The predictive performances, as assessed by the AUCs for the noradrenergic biochemical profile of PPGLs and found valid in a sham experiment, were not significantly different between radiomic features derived from VOIs with and without central necrosis. An explanation for this could be that the radiomic feature set describes many different types of heterogeneity and, as a result, feature sets from both delineation methods contain useful features in terms of predictive performance. Multicollinearity within one feature set is high, but might also be high between feature sets of different delineation methods. Additionally, the combination of radiomic features from both delineation methods resulted in an AUC similar to the ones of the different delineation methods seperately. This indicates that, in this small dataset, the combination of features from the two delineation methods does not result in additional information that is favorable for the predictive performance. It should be taken into account that, in order to prevent overfitting of the model, only three features could be retained in factor analysis. In other tumour types, multicollinearity in and between feature sets of different delineation methods is expected to be high as well, but further research is needed to confirm this effect in a larger population as well as for different tumour types.

This study showed that the effect of inclusion or exclusion of the region of central necrosis in the delineation significantly impacts radiomic feature values in PPGL and NSCLC, but does not impact the predictive performance of the PPGL radiomic model. A guideline on the choice to add or leave out central necrosis in delineation could not be provided. From a biological perspective, regions of central necrosis are part of the tumour and should therefore be included in the VOI, especially considering that central necrosis is associated with poor prognosis [6]. On the other hand, from a data-driven perspective, some features might already capture the presence of central necrosis without our awareness, since the features are investigated exploratively and without biological rationale. Both delineation methods can be used in radiomic studies, but feature values vary largely between both methods. For reproducibility purposes, especially in the setting of external validation [33], future studies should report whether regions of central necrosis were included in the delineation.

## 5. Conclusions

Central necrosis of tumours on [^18^F]FDG PET significantly impacts radiomic feature values. Almost two-third of the features were affected, demonstrating that the influence of whether or not to include regions of central necrosis in the delineation of the tumour on the radiomic feature values is significant. However, no significant difference in the predictive performance of both delineation methods was observed. In order to advance reproducibility of radiomic research, radiomic studies should report on whether or not central necrosis was (manually) included during delineation.

## Figures and Tables

**Figure 1 diagnostics-11-01296-f001:**
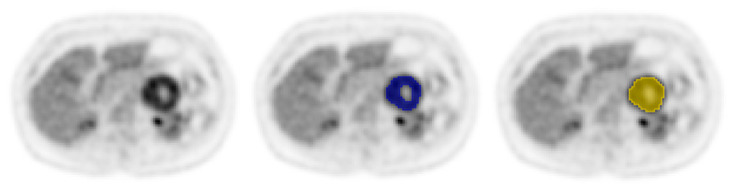
Example of a [^18^F]FDG PET/CT scan of a patient with a pheochromocytoma with central necrosis in the left adrenal gland, with VOI_vital-tumour_ in blue and VOI_gross-tumour_ in yellow. The NTF is 0.05.

**Figure 2 diagnostics-11-01296-f002:**
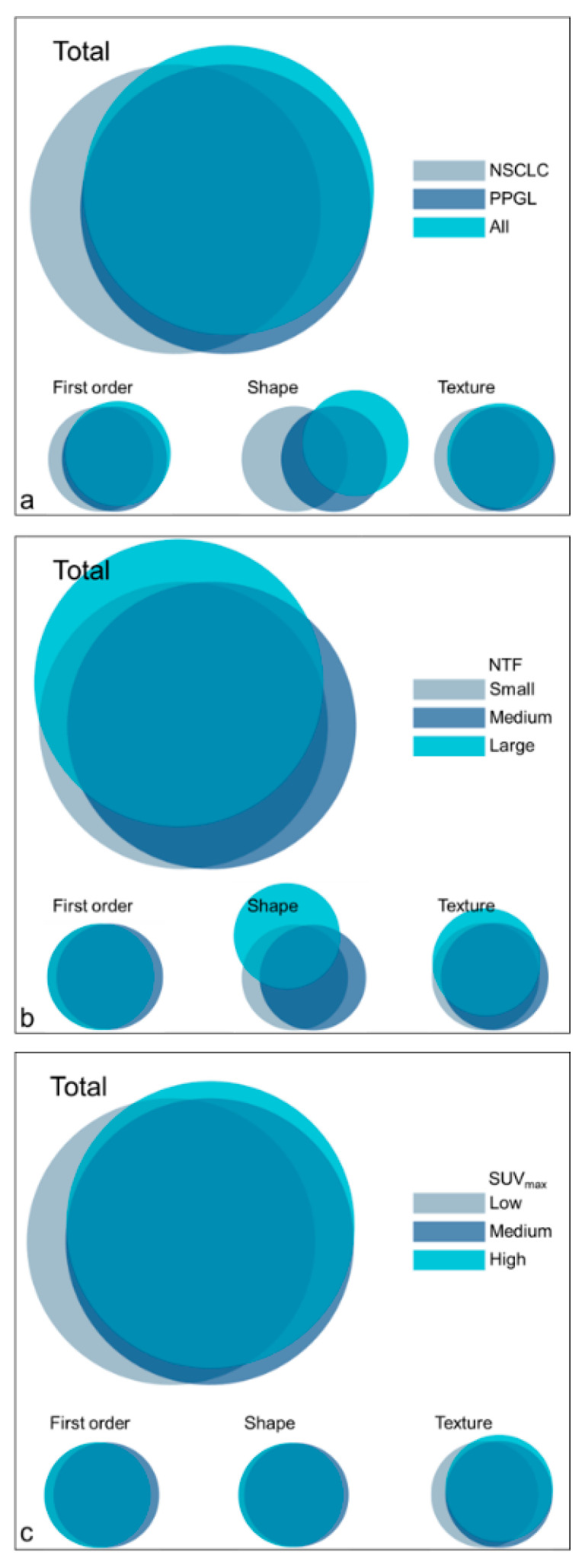
Venn diagrams representing the overlap in affected features per feature class for the cohorts (**a**) and subgroups ((**b**): NTF, (**c**): SUV_max_). NSCLC: non-small cell lung carcinoma, PPGL: pheochromocytoma and paraganglioma, NTF: necrotic tumour fraction, SUV_max_: maximum standardised uptake value.

**Figure 3 diagnostics-11-01296-f003:**
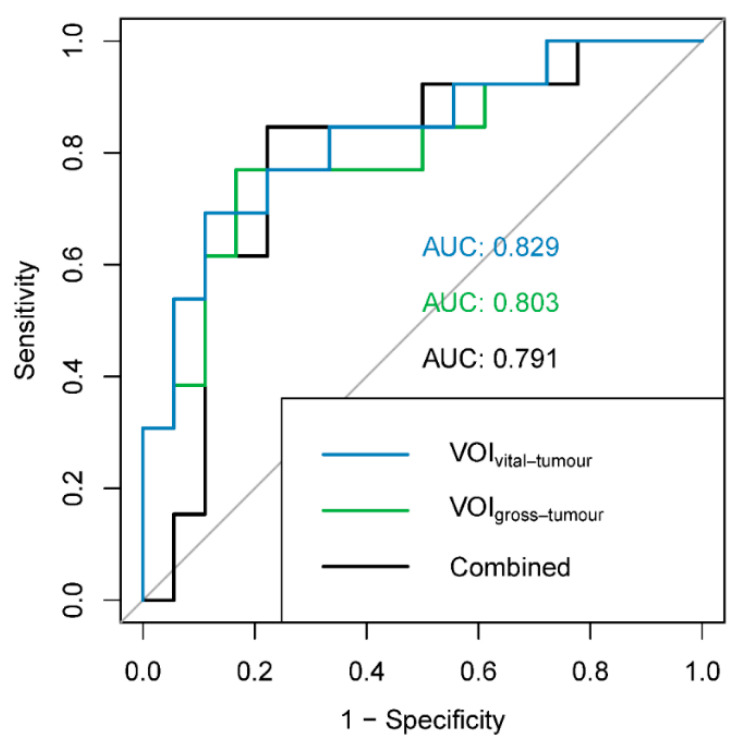
ROC curves and AUCs for the different radiomic models: blue: VOI_vital-tumour_ (features: first order minimum, shape surface area and GLCM informational measure of correlation 2); green: VOI_gross-tumour_ (features: shape surface area, NGTDM complexity and GLDM dependence entropy); black: combined (features: VOI_gross-tumour_ GLCM sum entropy, VOI_vital-tumour_ shape maximum 3D diameter and VOI_vital-tumour_ shape surface volume ratio). ROC: receiver operating characteristic, AUC: area under the ROC curve.

**Table 1 diagnostics-11-01296-t001:** Clinical characteristics of 31 PPGL and 12 NSCLC patients with central necrosis. SUV: standardised uptake value, MTV: metabolic tumour volume, NTF: necrotic tumour fraction, NSCLC: non-small cell lung carcinomas, PPGL: pheochromocytomas and paragangliomas.

	NSCLC (*n* = 12)	PPGL (*n* = 31)
Age (years), median (range)	65 (44–80)	62 (23–80)
Sex (M/F)	11/1	10/21
Histology	Adenocarcinoma: 4 Squamous cell carcinoma: 7 Other: 1	Pheochromocytoma: 26 Paraganglioma: 5
Noradrenergic biochemical profile (yes/no)	-	13/18
SUV_max_ (g/mL), median (range)	16.00 (9.05–29.77)	5.00 (2.54–36.01)
MTV_vital-tumour_ (cm^3^), median (range)	36.3 (15.2–173.4)	58.3 (16.6–388.8)
MTV_gross-tumour_ (cm^3^), median (range)	45.1 (18.7–327.3)	103.8 (28.7–611.2)
NTF, median (range)	0.19 (0.03–0.77)	0.23 (0.01–0.97)

**Table 2 diagnostics-11-01296-t002:** Numbers and percentages of features affected by the delineation method (VOI_vital-tumour_ vs. VOI_gross-tumour_) per feature class for the different cohorts and both cohorts together. Differences in the number of affected features between cohorts were assessed using the Fisher’s exact test. NSCLC: non-small cell lung carcinoma, PPGL: pheochromocytoma and paraganglioma, GLCM: grey level cooccurrence matrix, GLRLM: grey level run length matrix, GLSZM: grey level size zone matrix, GLDM: grey level dependence matrix, NGTDM: neighbouring grey tone difference matrix.

Feature Class	All (*n* = 43)	NSCLC (*n* = 12)	PPGL (*n* = 31)	*p*-Value
First order (18)	17 (94%)	13 (72%)	16 (89%)	0.402
Shape (14)	11 (79%)	7 (50%)	7 (50%)	1.000
Texture (73)	58 (79%)	48 (66%)	60 (82%)	0.024
GLCM (22)	15 (68%)	12 (55%)	15 (68%)	0.537
GLRLM (16)	15 (94%)	12 (75%)	15 (94%)	0.333
GLSZM (16)	12 (75%)	11 (69%)	13 (81%)	0.685
GLDM (14)	12 (86%)	9 (64%)	13 (93%)	0.165
NGTDM (5)	4 (80%)	4 (80%)	4 (80%)	1.000
Total (105)	86 (82%)	68 (65%)	83 (79%)	0.031

**Table 3 diagnostics-11-01296-t003:** Numbers and percentages of features affected by the delineation method (VOI_vital-tumour_ vs. VOI_gross-tumour_) per feature class for the subgroups based on NTF. Differences in the number of affected features between subgroups were assessed using the Fisher’s exact test. NTF: necrotic tumour fraction, GLCM: grey level cooccurrence matrix, GLRLM: grey level run length matrix, GLSZM: grey level size zone matrix, GLDM: grey level dependence matrix, NGTDM: neighbouring grey tone difference matrix.

Feature Class	All (*n* = 43)	Small NTF (NTF ≤ 0.12, *n* = 14)	Medium NTF (0.12 < NTF ≤ 0.36, *n* = 15)	Large NTF (NTF > 0.36, *n* = 14)	*p*-Value
First order (18)	17 (94%)	14 (78%)	16 (89%)	14 (78%)	0.745
Shape (14)	11 (79%)	5 (36%)	8 (57%)	10 (71%)	0.199
Texture (73)	58 (79%)	52 (71%)	60 (82%)	51 (70%)	0.170
GLCM (22)	15 (68%)	13 (59%)	15 (68%)	15 (68%)	0.850
GLRLM (16)	15 (94%)	12 (75%)	14 (88%)	14 (88%)	0.701
GLSZM (16)	12 (75%)	12 (75%)	15 (94%)	9 (94%)	0.058
GLDM (14)	12 (86%)	10 (71%)	11 (79%)	11 (79%)	1.000
NGTDM (5)	4 (80%)	5 (100%)	5 (100%)	2 (40%)	0.066
Total (105)	86 (82%)	71 (68%)	84 (80%)	75 (71%)	0.116

**Table 4 diagnostics-11-01296-t004:** Numbers and percentages of features affected by the delineation method (VOI_vital-tumour_ vs. VOI_gross-tumour_) per feature class for the different subgroups based on SUV_max_. Differences in the number of affected features between subgroups were assessed using the Fisher’s exact test. SUV_max_: maximum standardised uptake value, GLCM: grey level co-occurrence matrix, GLRLM: grey level run length matrix, GLSZM: grey level size zone matrix, GLDM: grey level dependence matrix, NGTDM: neighbouring grey tone difference matrix.

Feature Class	All (*n* = 43)	Low SUV_max_ (SUV_max_ ≤ 4.61, *n* = 14)	Medium SUV_max_ (4.61 < SUV_max_ ≤ 12.09, *n* = 15)	High SUV_max_ (SUV_max_ > 12.09, *n* = 14)	*p*-Value
First order (18)	17 (94%)	14 (78%)	15 (83%)	16 (89%)	0.898
Shape (14)	11 (79%)	4 (29%)	5 (36%)	4 (29%)	1.000
Texture (73)	58 (79%)	41 (56%)	49 (67%)	53 (73%)	0.106
GLCM (22)	15 (68%)	15 (68%)	12 (55%)	12 (55%)	0.610
GLRLM (16)	15 (94%)	9 (56%)	13 (81%)	15 (94%)	0.051
GLSZM (16)	12 (75%)	7 (44%)	11 (69%)	12 (75%)	0.249
GLDM (14)	12 (86%)	8 (57%)	9 (64%)	10 (71%)	0.919
NGTDM (5)	4 (80%)	2 (40%)	4 (80%)	4 (80%)	0.500
Total (105)	86 (82%)	59 (56%)	69 (66%)	73 (70%)	0.120

**Table 5 diagnostics-11-01296-t005:** Results of dimension reduction and predictive performance of the VOI_vital-tumour_, VOI_gross-tumour_ and combined model for the noradrenergic biochemical profile in the PPGL cohort. For each model, 3 features were selected corresponding to the factors with the highest loadings. AUCs of the radiomic models and the mean AUC of the sham experiment are reported. Features marked with * are not significantly different between delineation methods in the PPGL dataset. KMO: Kaiser–Meier–Olkin (KMO) measure, AUC: area under the receiver operating characteristic curve.

	VOI_vital-tumour_	VOI_gross-tumour_	Combined
Features retained after filtering (τ = 0.95)	53/105	57/105	103/210
KMO	0.970	0.969	0.986
Cumulative variance	0.63	0.58	0.53
Selected features	- First order Minimum - Shape Surface Area * - GLCM Informational Measure of Correlation 2	- Shape Surface Area * - NGTDM Complexity - GLDM Dependence Entropy	- VOI_gross-tumour_ GLCM Sum entropy - VOI_vital-tumour_ Shape maximum 3D diameter * - VOI_vital-tumour_ Shape Surface Volume ratio
AUC (95% CI)	0.829 (0.677–0.981)	0.803 (0.640–0.967)	0.791 (0.618–0.963)
Mean AUC sham experiment (100 iterations)	0.643	0.660	0.666

## Data Availability

The datasets generated during and/or analysed during the current study are available from the corresponding author upon reasonable request.

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
