# Peer review of "The Influence of the Exclusion of Central Necrosis on [18F]FDG PET Radiomic Analysis"

_diagnostics, 2021, doi:10.3390/diagnostics11071296_

Round 1

Reviewer 1 Report

The paper demonstrates that radiomic analysis performed by including the central necrosis gives different results than radiomic analysis without central necrosis. This is quite obvious but maybe there is the need to clearly state, as done by the authors, that radiomic studies should report whether or not central necrosis is included.

1) I am confused regarding the ROC analysis and I am not entirely sure about the validity of this test the way it has been used. What do the authors mean by “predictive performance”? Predictive of what? What outcome predictors have been tested against the three radiomic models of choice? I don’t see any patients’ outcome specified.

2) In addition, the authors should more clearly specify, both in the abstract and in the discussion section lines 277-279 and 371-372, that this ROC analysis was performed only on paragangioma patients.  Is it possible to generalize this result to other tumor types?

3) The authors selected three features for each of the three models (Table 5). Does this mean that the AUC is the average AUC across these three features? Please specify it clearly.

4) The authors should better justify their choice of including tumors with very different biology and outcomes in this analysis.

Author Response

We would like to thank both reviewers for their helpful comments, which helped us to improve the quality of the manuscript. All comments of the reviewers are provided below, together with our responses in italic. We have changed the manuscript according to these comments (track changes are marked in the manuscript). We hope we have sufficiently addressed all of the reviewers concerns below.

Reviewer 1

The paper demonstrates that radiomic analysis performed by including the central necrosis gives different results than radiomic analysis without central necrosis. This is quite obvious but maybe there is the need to clearly state, as done by the authors, that radiomic studies should report whether or not central necrosis is included.

Thank you for pointing out the importance of choice of delineation and we agree that this message should be clear. Some studies manually add the excluded regions of low tracer uptake to the volume of interest, but this is not always clearly reported. This is added to lines 60-62 of the introduction.

1) I am confused regarding the ROC analysis and I am not entirely sure about the validity of this test the way it has been used. What do the authors mean by “predictive performance”? Predictive of what? What outcome predictors have been tested against the three radiomic models of choice? I don’t see any patients’ outcome specified.

We apologize that this was not clearly stated. By ‘predictive performance’ we mean the predictive performance of the radiomic signature for underlying tumour biology. This was added to the introduction, line 76. From a methodological perspective, it is important to know which radiomic features change with the addition of central necrosis, but from a clinical perspective, it is much more important to know if the predictive performance changes. Therefore, the radiomic models have been tested against the noradrenergic biochemical profile. The methodology was clarified by specifically outlining, underlying tumour biology and the noradrenergic biochemical profile as response variable in a separate sentence (lines 172 and 177).

2) In addition, the authors should more clearly specify, both in the abstract and in the discussion section lines 277-279 and 371-372, that this ROC analysis was performed only on paragangioma patients.  Is it possible to generalize this result to other tumor types?

Thank you for pointing this out. We added that the predictive performance was only assessed in the PPGL cohort, both in the abstract (line 27) and in the discussion (lines 287, 367 and 386).

Considering the generalizability to other tumour types, although both tumour types have a distinctly different tumour-to-background ratio, we see that the features that are significantly different between delineation methods were highly similar between NSCLC and PPGL. These results are shown in line 253 and supplementary table S2, and described in the discussion, lines 358-362. We agree and added that this phenomenon could be generalized to other tumour types, but the number of affected features might vary as a result of tumour characteristics such as tracer uptake and distribution, tumour geometry and necrotic tumour fraction (added to lines 362-365). In terms of the predictive performance, we are not sure if the results can be generalized to other tumour types. We attribute the similar predictive performance between delineation methods to high multicollinearity in the feature set of one delineation method and between the feature sets of different delineation methods (lines 371-379). This is also probable for other tumour types, but was not assessed in our study and warrants for future research, as now described in the discussion, lines 381-383.

3) The authors selected three features for each of the three models (Table 5). Does this mean that the AUC is the average AUC across these three features? Please specify it clearly.

We agree with the reviewer that this should be described more clearly. The reported AUCs represent the models of multiple binary logistic regression with the three selected radiomic features as independent variables. Terminology was adapted to ‘multiple binary logistic regression’ and ‘radiomic model’ (methods line 196, figure 3, table 5). AUCs for individual features were not calculated.

4) The authors should better justify their choice of including tumors with very different biology and outcomes in this analysis.

The reviewer is right, this should be more clearly stated. Our methodology focuses on the effect of phenomenon of central necrosis on the radiomic analysis. Central necrosis often occurs in both NSCLC and PPGL. However, both tumors show different tumour characteristics (high tumour-to-background ratio in NSCLC vs low in PPGL). This provided us with the opportunity to study the effect of the inclusion of central necrosis on the feature values for different tumour characteristics (added to methods line 87-92). Association of radiomic features with outcome measures was only performed for the PPGL cohort.

Reviewer 2 Report

The work proposed by Noortman et al analyzed the differences in radiomic features in two cohorts with different tumor types to determine whether the inclusion of central necrosis in tumor delineation is an important factor to consider when performing clinical radiomic studies. The paper is exhaustive and well organized: it analyzes the topic in a complete way, highlighting the results as well as limitations in a clear and complete way.
For the above reasons, I recommend the publication of the paper in the proposed version.

Author Response

We would like to thank both reviewers for their helpful comments, which helped us to improve the quality of the manuscript. All comments of the reviewers are provided below, together with our responses in italic. We have changed the manuscript according to these comments (track changes are marked in the manuscript). We hope we have sufficiently addressed all of the reviewers concerns below.

The work proposed by Noortman et al analyzed the differences in radiomic features in two cohorts with different tumor types to determine whether the inclusion of central necrosis in tumor delineation is an important factor to consider when performing clinical radiomic studies. The paper is exhaustive and well organized: it analyzes the topic in a complete way, highlighting the results as well as limitations in a clear and complete way.
For the above reasons, I recommend the publication of the paper in the proposed version.

We appreciate and would like to thank the reviewer for the positive feedback.

Round 2

Reviewer 1 Report

The authors have addressed my comments in sufficient detail.